# The Association between Women’s History of Sexual Abuse, Mental Health, and Sexual Motivations in Committed Intimate Relationships

**DOI:** 10.3390/healthcare12030389

**Published:** 2024-02-02

**Authors:** Roi Estlein, Ronit Smadar Dror, Zeev Winstok

**Affiliations:** School of Social Work, University of Haifa, Haifa 31905, Israel; ronitsd@netvision.net.il (R.S.D.); zeevwin@research.haifa.ac.il (Z.W.)

**Keywords:** mental health, depression, women’s sexual abuse, sexual motivations, intimate relationships

## Abstract

In committed intimate relationships, motivations for engaging in or avoiding sexual relations can indicate partners’ perceptions, needs, and attitudes toward sexual intimacy, and reflect sexual functioning. Sexual motivations can be positive, reflecting and advancing relational goals, such as establishing and maintaining closeness between partners and pleasure, or negative, stemming out of fear of one’s partner, pleasing them, or depriving sexual contact to punish the partner or establish relational power. In this study, we extended the current conceptualization and assessment of negative sexual motivations to explore the associations between women’s history of sexual abuse, their mental health, and their negative sexual motivations. Structural equation modeling results from 236 adult Israeli women who were in committed intimate relationships indicated that a woman’s history of sexual abuse negatively predicted her mental health which, in turn, negatively predicted her negative sexual motivations. Mental health mediated the association between a woman’s history of sexual abuse and her negative sexual motivations. These findings have theoretical and empirical contributions to research in terms of the short- and long-term effects of sexual abuse on women, mental health, and women’s sexuality. Their clinical implications for mental health professionals, sexual therapists, and clinicians working with women who experience sexual abuse are also discussed.

## 1. Introduction

According to the World Health Organization [1], mental health refers to the extent to which individuals assess and understand their personal abilities, and are able to productively and effectively cope with everyday stresses and contribute to their community. Among the numerous indicators of mental health, depression and self-esteem are the most significant. Whereas *depression* is an emotional aspect of mental health, *self-esteem* constitutes a cognitive aspect [2]. Prior research has consistently documented negative associations between the two aspects [3,4]. There are numerous factors that can contribute to women’s depression and self-esteem as markers of their mental health [5,6]. One’s history of sexual abuse is one such significant factor, and can have deleterious implications for a woman’s mental health [7]. In the current study, we highlight depression and self-esteem as markers of mental health among women and their association with their history of sexual abuse, both in childhood and adulthood.

Additionally, we examined mental health as a predictor of sexual motivations among women, including fear-based (e.g., “I was afraid to say ‘no’ to him”) or pleasing (e.g., “because he needed it”), and depriving sex for the purpose of establishing power in the relationship (“so I can feel stronger than him”) or punishing a partner (“to punish him”). These sexual motivations are often considered negative because, contrary to other motivations whose goal is to establish and promote intimacy and joint pleasure, which the literature refers to as positive and desirable [8,9], these motivations reflect the opposite. Accordingly, we also explored the possibility that mental health (i.e., depression and self-esteem) mediates the association between women’s history of sexual abuse and their sexual motivations.

### 1.1. Mental Health

*Depression* refers to mood disturbance characterized by loss of pleasure and the absence of positive affect, which has consequences for the individual in terms of functioning [10,11]. Whereas depression is often viewed as either pathological or non-pathological, other approaches consider depression as a continuum, varying from mild forms to severe depressive illness [12,13,14,15,16]. In its pathological state, depression is quite common. In Israel and the United States, for example, nearly one in ten adults reported a depressive mood disorder over the passing year, and one in six reported experiencing depressive symptoms sometime during life [17,18]. In addition, women seem to be at a higher risk for depression than men [14,15,19].

Studies have identified various predictors of depression, including genetic, neurobiological, cognitive, and environmental factors [20]. There is evidence pointing to associations between aversive life events and later depression [21,22,23,24]. An example is the ever-growing body of research on the links between history of sexual abuse in childhood and depression: sexual history of abuse can have a long-term effect on depression among women, even years later [25,26]. Depression also has implications on one’s daily functioning, including sexual functioning. Indeed, studies point to associations between high levels of depression and different aspects of sexual dysfunction [27,28]; among women, depression predominantly predicts decreases in sexual desire, and among men, both sexual desire and erectile function tend to be impaired [29].

*Self-esteem* represents the subjective perception of individuals about their value, which does not necessarily reflect their abilities or objective achievements [30,31,32]. Self-esteem is a cognitive schema consisting of two main components, namely content and structure [33]. The content component encompasses knowledge aspects (e.g., “who I am”, “what I am”) and evaluative aspects (e.g., “how I feel about myself”) that range on a spectrum of positive perceptions (e.g., “I am a success”) on one extreme, to negative perceptions (e.g., “I am a failure”) on the other [30,33]. The structure component refers to the way information and convictions about oneself are organized (Campbell et al., 1996 [33]), which can be general or more context related (e.g., professional, social, or intimate contexts; Leary and Baumeister, 2000 [30]). Some research has explored self-esteem within a sexual context (e.g., Snell et al., 1992 [34]); however, most of that research assessed participants’ global evaluation of their self-esteem [31], although studies suggest it can change over the years [35] such that it tends to increase during young adulthood and in midlife, then reaches its peak at age 60–70, and then sharply decreases in late adulthood [36]. Studies also documented gender differences in self-esteem, such that men tend to report higher self-esteem than women in terms of their looks and physical abilities, and women often report higher self-esteem than men in terms of social acceptance and ethics [37].

Like depression, self-esteem is influenced by life experiences and events [31]. A history of sexual abuse in childhood can be a significant life event that may be associated with self-esteem. Indeed, such abuse can have a negative long-term effect on self-esteem, even years later [38,39]. Also, like depression, self-esteem can have implications on one’s daily functioning, including their sexual functioning. Low self-esteem can interfere with one’s need for sexual intimacy, ability to love, and maintain an intimate relationship [40].

### 1.2. Sexual Motivations

Motivations for engaging in or avoiding sexual relations in intimate partnerships can constitute important indicators or characteristics of one’s sexual functioning. Such motivations integrate the characteristics of sexual functioning, including one’s perceptions, needs, behavioral patterns, and their experience of sexual activities. Research on sexual motivations differentiates between positive and negative motivations (i.e., reasons) for having sex [8,9]. Whereas positive motivations are considered constructive for the relationship and help to establish and maintain intimacy (e.g., “I had sex with partner to show them I love them”) and pleasure (e.g., “I had sex with my partner to satisfy my sexual needs”), negative sexual motivations are considered destructive for the relationship, and include social pressure (e.g., “I had sex because I was afraid he/she would think negatively about me if I didn’t”), coping with negative feelings (e.g., “I had sex to reduce my loneliness”), a sense of insecurity (e.g., “I had sex because I feared that my partner would not love me anymore”), and self-approval (e.g., “I had sex to prove to myself that I’m attractive”).

Recently, two more sexual motivations considered negative were identified. One is an extension of the insecurity motivation [8], and includes motivation for sex either out of fear of one’s partner (e.g., “because I was afraid to refuse him”) or in order to please one’s partner (e.g., “because he needed it”). The other motivation refers to sexual avoidance, which is motivated by the need to establish status and power in the relationship (e.g., “so I feel I am stronger than him”) and to punish one’s partner (e.g., “to make him suffer”). In the current study, these two motivations are considered indicators of sexual dysfunction, which can be associated with poor mental health.

### 1.3. Sexual Abuse

Sexual abuse is widely defined as coerced sexual contact against the will of another person [41]. It varies in terms of the type of the sexual encounter or act (e.g., forced touch, exposure to exhibitionists), and involves threatening [42]. Sexual abuse can occur in childhood as well as in adulthood, with increased prevalence rates that are said to range from 4% to 21% in adults, and from 5% to 33% in children in the general population [43]. The high numbers of women who have experienced sexual abuse during their lives have drawn the attention of researchers, and motivated them to explore the associations between women’s history of abuse and psychiatric and medical symptoms [44]. Although this research has yielded significant information on the personal implications of one’s history of sexual abuse, less is known about the implications it may have for interpersonal and sexual characteristics in intimate relationships, especially in terms of women’s sexual motivations.

### 1.4. The Current Study

This study explores the associations between a history of sexual abuse, mental health (i.e., depression and self-esteem), and negative sexual motivations in the form of fear of and pleasing a partner, and establishing power or punishing one’s partner among women in committed intimate relationships. The current study contributes to the literature in several ways. Firstly, most research nominates only depression as an indicator of mental health. Self-esteem is much less studied, and there are few studies that employed both depression and self-esteem. Secondly, previous studies separately examined the associations between the variables in our model, but the relationship between both of them is unknown. Finally, this is the first time that sexual motivations have been used to assess sexual function. Thus, based on the reviewed literature, we hypothesize the following:

**Hypothesis** **1** **(H1).**
*A woman’s history of sexual abuse will be negatively associated with her positive mental health, such that the higher her score is for sexual abuse, the lower the level of her mental health will be (i.e., high depression and low self-esteem).*


**Hypothesis** **2** **(H2).**
*A woman’s mental health will be negatively associated with her negative sexual motivation of fear and pleasing, establishing power, or punishing her partner.*


## 2. Materials and Methods

### 2.1. Procedure

We conducted an online survey using a convenience sample of women in Israel. Participants were recruited via social media (Facebook and Instagram). The survey was accessible through Qualtrics, a secure web-based survey data collection system. The survey took 15 min to complete, on average, and was open from January to February 2023. The survey was anonymous, and no data were collected that linked participants to recruitment sources. The University of Haifa’s institutional review board (IRB) approved all procedures and instruments. The inclusion criteria were adult women who were engaged in a heterosexual relationship for at least the last 12 months, and who had sexual relations with their partner. These women were invited to participate in a study on mental health and intimate partner sexual relationships, by clicking on the survey link. Clicking on the link guided potential respondents to a page with information about the purpose of the study, the nature of the questions, and a consent form (i.e., the survey was voluntary; respondents could skip questions or quit at any time; responses would be anonymous). In addition, the first page offered contact information of the researchers, who are social workers, should participants feel distress or the need for assistance. No compensation was given for participating in the study.

### 2.2. Participants

The sample consisted of 236 adult Israeli women who were in a committed relationship with a partner for at least 12 months, and with whom they had sexual relations. The sample size was a priori, as determined by a power analysis using GPower* 3, in order to reach 95% power to detect medium 0.15-sized effects. The inclusion criteria were that participants had to be women, and were engaged in a romantic committed intimate partnership for at least 12 months. The participants’ ages ranged from 18 to 68 years (*M* = 27.98, *SD* = 7.93). Most of the women sampled were Jewish (98.3%) and were born in Israel (91.5%). Most women defined themselves as non-religious or secular with traditional orientation (79.6%), while the remainder identified as traditional (10.6%), religious, or ultra-orthodox (9.8%). All of the women in the sample reported being in a relationship. Their mean relationship tenure was 5.59 years (*SD* = 6.22). Some were married (30.5%), others cohabited with their partner without being married (38.6%), and others did not live with their partner under the same roof (30.9%). Importantly, all participants defined their relations with their partner committing and exclusive. Most of the women reported an average economic status (66.9%), while fewer reported a lower economic status (28.4%), and a smaller percentage reported a higher economic status (4.7%). 

### 2.3. Measuring Instruments

#### 2.3.1. Self-Esteem

We employed the widely used Rosenberg’s [45] Self-Esteem Scale. This measurement consists of 10 statements, with 5 positive items (e.g., “I feel that I’m a person of worth, at least on an equal plane with others”), and 5 negative items (e.g., “I certainly feel useless at times”). The response options were the following: (1) strongly disagree; (2) disagree; (3) agree; and (4) strongly agree (*Cronbach’s α* = 0.87). According to these results, the research variable was computed based on mean positive statements and inverse negative statements (*M* = 3.10, *SE* = 0.03).

#### 2.3.2. Depression

We used Bech et al.’s [46,47] Major Depression Inventory (MDI), which consists of 12 items asking the research participants to report how often they have experienced negative emotions (e.g., low mood, scruples and guilt, difficulty to concentrate, restlessness) within the preceding two-week timeframe. The response options were the following: (1) never; (2) a little bit of the time; (3) some of the time; (4) a lot of the time; (5) most of the time; (6) all the time (*Cronbach’s α* = 0.86). According to these results, a research variable was computed based on the average answer score (*M* = 2.46, *SE* = 0.05). The correlation between depression and self-esteem items average was examined. As expected, a strong significant correlation was found (*r_p_* = −0.56).

#### 2.3.3. Compliance in Sex Scale (CSS) and Withholding Sex Scale (WSS)

The Compliance in Sex Scale (CSS) and Withholding Sex Scale (WSS) were developed in a previous study [48,49] whose main purpose was to test the reliability and validity of the scales. The test was conducted using a sample of 675 men and women who were engaged in an intimate sexual relationship for the past 12 months or longer. The validity examination of the two measures was based on face validity, as evaluated by experts in the field of sexuality and intimate relationships, in terms of items distribution, the factorial structure of the measurement as part of its structural validation, various approaches for different purposes in variable encoding, the relationships between measurement components, as well as between them and sex life quality, and as part of criterion validity. The findings supported the validity of the measurements, and their reliability was high (for CSS: *Cronbach’s α* = 0.93, 15 items; for WSS: *Cronbach’s α* = 0.92, 12 items).

Specifically, the Compliance in Sex Scale (CSS) assesses two interrelated aspects of sexual compliance: (1) engaging in sex to please the other (pleasing factor), and (2) engaging in sex out of fear of the consequences of refusing to do so (fear factor). The principles of the measurement are consistent with those underlying the sexual motives measure by Cooper et al. [8]. Like Cooper’s measure, this scale examines behaviors driven by motivations. Cooper’s measuring instrument includes a sub-measure of sex motivated by insecurity, using four items (items 8, 13, 26, 28 in the original questionnaire), but does not cover compliance in sex as a whole because insecurity is perceived as being a part of pleasing and fearing the other in sexual behavior. Accordingly, Cooper’s four insecurity items were restructured in the present measure, with an additional eleven items to expand the measure of compliance in sex.

The following introduction to the questionnaire was presented to the research participants: “Sometimes, one partner would have sex with the other partner even if they don’t feel like it. Consenting to sexual relations without desire or inclination may stem from various reasons, sometimes several all at once. This part of the questionnaire only refers to the times you had sex with your partner unwillingly, that is, if it was only up to you, you would not have engaged in sexual relations with your partner”. The guiding question to each questionnaire item was the following: “Over the last 12 months, of all the times you unwillingly agreed to have sex with your partner, it was for the following reasons (each case can have several reasons)”. Each item that followed stated a reason for having sex. An example of a questionnaire item expressing fear as the reason for sexual intercourse is the following: “Because I was afraid that my partner would leave me”. An example of pleasing as the reason for sex is as follows: “To please my partner”. Each of the items was rated on a 6-point scale as follows: 0—never happened during the last 12 months or before; 1—never happened during the last 12 months only; 2—happened rarely; 3—almost never happened; 4—happened almost every time my partner wanted to; 5—happened often but not always. The measurement reliability in the present study was high (internal consistency using *Cronbach’s α* = 0.92, 15 items). A new variable was computed based on the mean answer score (*M* = 0.95, *SE* = 0.05).

In terms of the Withholding Sex Scale (WSS), the scale was developed to measure two interrelated aspects of withholding sex: withholding sex to punish the partner (punishment factor) and withholding sex to gain power over the partner (power factor). Here, too, the principles of the measurement were consistent with those underlying the sexual motives measure by Cooper et al. [8]. Like Cooper’s measure, this scale examines behaviors driven by motivations. The measure consisted of 12 Items.

The following introduction to the questionnaire was presented to the research participants: “Sometimes, one partner wants to have sex but the other partner refuses. The refusal may stem from various reasons, and even more than one. This part of the questionnaire addresses only some of the reasons to avoid sexual relations with a partner. It addresses the times you could accept your partner’s request for sex, but you refused. The questions do not refer to the times you avoided having sex due to incapability, physical, or mental limitation, or because you were disinterested, unattracted or repulsed”. The guiding question to the questionnaire items was the following: “Over the past 12 months, out of all the times your partner wanted to have sex with you, and you could agree to it but refused, it happened for the following reasons (each time can have several reasons)”. Each item that was stated following the question included a reason for not engaging in sexual intercourse. An example of an item expressing sexual avoidance as penalty is as follows: “To teach him/her a lesson”. An example of an item expressing sexual avoidance to gain power is the following: “To show that I am the one in control here”. Each of the items was rated on a 6-point scale as follows: 0—never happened in the last 12 months or before; 1—never happened during the last 12 months only; 2—happened rarely; 3—almost never happened; 4—happened almost every time my partner wanted to; 5—happened from time to time but not always. The measurement reliability was high (internal consistency using *Cronbach’s α* = 0.94, 12 items). A new variable was computed based on the average answer score (*M* = 0.54, *SE* = 0.05). The correlation between withholding sex and compliance in sex items average was examined. As expected, a strong significant correlation was found (*r_p_* = 0.54), indicating a shared factor for both measurements.

#### 2.3.4. Sexual Abuse

Sexual abuse was measured using the Sexual Abuse History Questionnaire (SAHQ) [50]. This questionnaire consists of six various sexual abuse items, differentiating between childhood (13 years and younger) and adulthood (14 years and older). The following introduction to the questionnaire was presented to the participants: “Women can experience unwanted events of a sexual nature as girls 13-year-old or younger, and as grown-ups of 14 years or older. Please try to recall whether such events happened to you as a child or as an adult”. An example of a questionnaire item is as follows: “someone ever exposed their genitals in front of you without you wanting it”. Another example is the following: “someone ever made you have sexual relations without you wanting it”. Each item had the following response options: 0—it did not happen to me as a child or an adult; 1—it happened to me only as an adult; 2—it happened to me only as a child; 3—it happened to me both as a child and as an adult. The research variable was computed in the following manner: in the first step, the 6 items were recoded into 12 dichotomous items differentiating between “it happened” (1) and “it did not happen” (0), and between childhood sexual abuse (6 items) and adulthood sexual abuse (6 items). The measurement reliability was acceptable (internal consistency using *Cronbach’s α* = 0.75, 12 recoded items). In the second step, the 12 items were summed up to arithmetically represent a history of sexual abuse.

### 2.4. Analytic Procedure

The first step was to test measurement reliability, encoding, and computing the research variables accordingly. The second step was to test the research hypotheses using structural equation modeling (SEM). First, the correlations between variables were examined prior to testing the research model. Then, the research model was tested to allow research hypotheses examination. The research model was tested using AMOS-27 software.

## 3. Results

A preliminary step, before testing the hypotheses, was to examine the research variables’ correlations and means. The findings of this analysis are presented in Table 1.

Next, the research hypotheses were tested using a model consisting of two latent factors and three control variables. The first latent factor represented mental health using two indicators: depression and self-esteem. The second factor represented sexual motivations considered negative, using two indicators as well: consenting to unwanted sex, and withholding sex. Mental health was presented in the model as the factor affecting sexual motivations. The effect of three additional variables, including sexual abuse history, age, and education on the latent factors, was also examined. Figure 1 presents the analysis findings. The model yielded good fit indexes (χ2_(9)_ = 15.024 *p* < 0.05, NFI = 0.95, IFI = 0.98, CFI = 0.98, RMSEA = 0.053, N = 236). The latent factor representing mental health loaded positively on the indicator representing self-esteem (*β* = 0.72), and negatively on the indicator representing depression (*β* = −0.78), suggesting it represents a tendency toward positive mental health. The latent factor representing the negative sexual motivations loaded positively on the indicator representing consenting to unwanted sex (*β* = −0.76), and negatively on the indicator representing withholding sex (*β* = 0.71), suggesting it represents a tendency toward negative motivations. Subject to the research hypotheses, it was found that the positive mental health factor negatively affected negative sexual motivations (*β* = −0.64). That is, the lower the positive mental health, the higher the negative sexual motivations. In greater detail, the stronger the tendency toward depression, and the lower the tendency toward self-esteem, the higher the tendency toward consenting to unwanted sex and toward withholding sex. Further analysis of the model indicated that the mental health factor was positively affected by the age (*β* = 0.28) and education (*β* = 0.20) of the participants, and negatively affected by their sexual abuse history (*β* = −0.22). The negative sex motivations factor was positively affected only by the participants’ age (*β* = 0.38). The participants’ sexual abuse history and education had no significant effect on this factor. Thus, the findings supported the main research hypotheses. A negative association was found between the childhood and adulthood sexual abuse history and positive mental health (*β* = 0.22). A negative association was also found between positive mental health and negative sex motivations (*β* = −0.64).

Finally, because the model found no direct effect of sexual abuse history on sexual motivations, only on mental health, which in turn affected sex motivations, it was explored whether the effect of sexual abuse history on motivations was mediated by mental health. Indeed, this examination provided support for the mediating role of mental health: when the effect of mental health was constrained to “0”, the effect of one’s history of sexual abuse on motivations turned out to be significant (*β* = 0.14, *p* < 0.05).

## 4. Discussion

This study focused on sexual motivations of women in committed intimate relationships. Our findings showed that mental health mediated the association between a woman’s sexual abuse history and her current sexual motivations. These findings address and integrate three research areas—namely, women’s sexual abuse, mental health, and women’s sexuality—and have theoretical and clinical implications for all three.

We employed two indicators of mental health in this study, depression and self-esteem. Whereas depression primarily represents an emotional aspect of metal health, self-esteem represents a cognitive aspect [2]. Previous studies documented a negative association between these two indicators [3,4], and our results showed that each was strongly associated with mental health. Our results also pointed to a negative effect of a woman’s sexual abuse history on her mental health, suggesting both short- and long-term effects on her well-being. In addition, we found an effect of mental health on women’s sexual motivations. Specifically, the more positive one’s mental health, the less negative sexual motivations she reported, and vice versa. Finally, a woman’s history of sexual abuse has an effect on sexual motivations only when mediated by mental health. These findings support our research hypotheses.

Like mental health, negative sexual motivations were represented by two indicators in this study, one was fearing of one’ partner and pleasing him, and the other was establishing power in the relationship and punishing the partner. Although these indicators may seem to be opposites (i.e., whereas the first suggests engaging in sexual relations out of weakness and self-depreciation, the other reflects attempts to take control and relational dominance), both reflect and create struggle rather than promote intimacy and closeness between partners. In this sense, this study offers three theoretical innovations: firstly, it nominates and explores two motivations that have received little attention in sexuality literature. Secondly, it is the first to conceptually integrate these motivations. Finally, it offers a new, non-judgmental perspective on these motivations. As indicated by our findings, there seems to be a strong positive association between these motivations, albeit with alleged contrast between them. We believe that this is because both motivations reflect one’s perception of sexual relations as a chore and/or a means to gain some kind of benefit, rather than create and maintain intimacy and pleasure. The positive correlation between these indicators, and their shared positive loading on the latent factor that represented negative sexual motivations in particular, strongly support this interpretation.

There is vast agreement that mental health and productive positive functioning are closely linked [1]. The current findings pointed to a strong negative association between women’s mental health and their sexual motivations, which constitute an important part of their sexual functioning. This association highlights the central role of mental health in women’s sexual functioning and probably, the central role of sexual functioning in women’s mental health. The negative direction of the documented association also implies the negative nature (i.e., not promoting intimacy and closeness) of these motivations. However, it is possible that in some cases these motivations can increase one’s self-esteem and reduce depression, especially among narcissistic individuals who may feel more positively about themselves when they establish control in their relationships [51]. Such possibilities call for caution when interpreting the valence of these motivations.

Our research model suggests that sexual motivations are the result of mental health. Importantly, this is only one way to interpret the associations between the study variables. Another interpretation would suggest that mental health is represented not only by depression and self-esteem but also by sexual functioning, including sexual motivations. In other words, it could be that sexual motivations—similar to depression and self-esteem—serve as an indicator of mental health rather than its consequence. The strong effect of mental health on sexual motivations in this study is quite similar to the effect received for the effect of mental health on depression and self-esteem. Moreover, the assessment of mental health by measuring depression and self-esteem is based on indicators of perception, meaning, experience, and functioning in different contexts [45,47,52], which is quite similar to how sexual motivations are measured. Therefore, the paths from mental health to sexual motivations in our model may be similarly considered as the paths from mental health to depression and self-esteem.

Studies have constantly showed that adverse life events can significantly impact individuals’ mental health and psychological well-being [2]. Our research model included participants’ sexual abuse history, which has been identified as a traumatic factor that can negatively impact mental health [53]. Since sexual abuse experiences are harmful and undesired, they can create negative perceptions about sexual relations. This makes history of sexual abuse particularly relevant to the current study, which explored not only mental health, but also sexual motivations: past negative sexual experiences can have destructive implications for various life functions, and especially for one’s sexual functioning and their motivations for sexual relations. Our findings highlight the negative effect of a woman’s history of sexual abuse on her current mental health, and its indirect effect on developing sexual motivations that do not promote constructive positive relational aspects, such as intimacy and pleasure; thus, its effect is considered negative. It appears that a history of sexual abuse may advance perceptions of sexual relations as a chore or a means to manipulate relational power in intimate relationships.

Although our results indicate an association between a women’s history of sexual abuse and her current mental health which, in turn, has an effect on her sexual motivations, it should be noted that this association is relatively mild. Accordingly, it is important to consider the possibility that women with a history of sexual abuse may not establish committed intimate relationships, while others who do may avoid sexual relations with their partner. Such women did not participate in the current study, which only included women engaged in committed romantic relationships with sexual relations with their partner. Importantly, the women who participated in this study seem to have coped, at least to an extent, with their past traumatic experiences, and were able to create an intimate sexual relationship with a partner.

### 4.1. Limitations and Future Directions

This study has significant conceptual, theoretical, and empirical strengths, but also has limitations that should be acknowledged. Firstly, the sample in this study was relatively small, and included mostly middle-class women who all have committed intimate relationships. It did not include men, less liberal and more traditional female participants, nor women with a history of sexual abuse who do not have a committed romantic relationship with an intimate partner. A larger more diverse sample in terms of gender, cultural and religious differences, and clinical characteristics can provide a deeper comparative understanding of the explored associations, and should be studied in future research. A dyadic approach to explore similarities and dissimilarities among partners [54] can also extend our understanding of the interdependence that exists in sexual and mental health processes. Secondly, some of the employed instruments for measuring the variables in this study may be somewhat narrow. For example, the measure of a woman’s history of sexual abuse may have missed some of the concept’s aspects, as it did not consider the characteristics of the abuse, such as who the abuser was, the severity of the abuse, and its frequency. Also, we did not assess participants’ resiliency, social support sources, and whether or not they had had therapy. In addition, the measures of mental health were not specific to sexual abuse. For example, it may have been relevant to include a measure of post-traumatic stress symptoms in addition to the depression and self-esteem measures. Finally, we only focused on negative sexual motivations that represent the half-empty glass of the studied phenomenon. It may have been beneficial to also examine more positive sexual motivations that do promote intimacy and closeness in the relationship, to also assess the half-full glass of sexual processes among women with a history of sexual abuse. In this sense, prior research has indicated that some women who had experienced sexual abuse report satisfying sexual relations [55,56], suggesting that the crucial aspect of healthy sexuality in intimate relationships is one that women with a history of sexual abuse are willing and are able to establish [57]. Expanding the pool of measures employed in the current study will help to portray a fuller picture and further understand the examined subject.

### 4.2. Clinical Implications

The findings of this study can inform mental health professionals, sexual therapists, and clinicians who work with women who have experienced sexual abuse. Although each group of professionals may focus their intervention on a different aspect of the current findings, they should all consider sexual abuse as a traumatic factor that has a significant direct effect on the mental health of women, and indirectly impacts their sexual intimate relationships. More specifically, mental health professionals should consider sexual motivations as an indicator of mental health, similar to depression and self-esteem. Sexual therapists should not necessarily interpret sexual motivations that do not promote intimacy and pleasure as negative, but rather examine such motivations and their implications on different individuals. Finally, clinicians working with women with a history of sexual abuse who are engaged in committed intimate relationships should take into consideration the possibility that the implications of sexual abuse experiences may indirectly influence sexual motivations and functioning, and thus may be less overt and only implied or elusive.

## 5. Conclusions

In this study, we explored the associations between women’s history of sexual abuse, their mental health, and their negative sexual motivations. Our findings highlight the negative effect of a woman’s history of sexual abuse on her current mental health in the form of depression and self-esteem, and its indirect effect on developing sexual motivations that do not promote constructive positive relational aspects, but rather negative ones. This study offers an extension of the conceptualization and evaluation of negative sexual motivations. In addition to its conceptual, theoretical, and empirical contributions, this study also highlights clinical implications of sexual abuse on women’s mental health and intimate relations, calling for further research to advance our understanding of the explored phenomenon.

## Figures and Tables

**Figure 1 healthcare-12-00389-f001:**
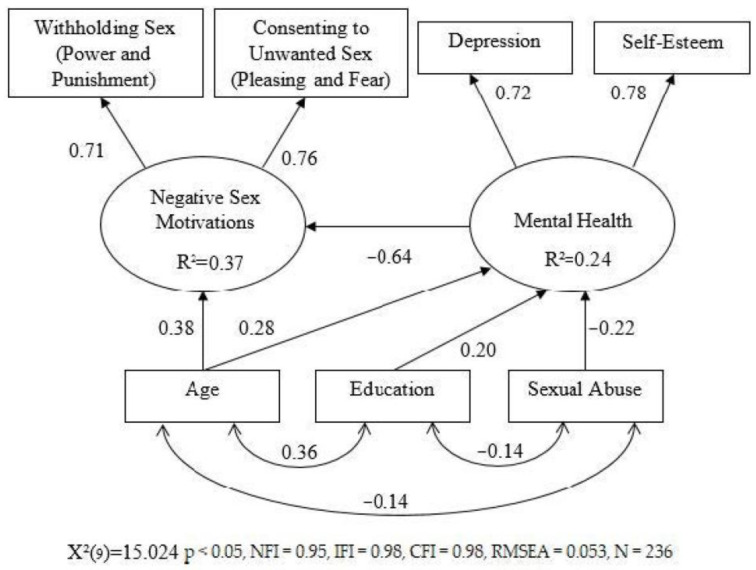
SEM model.

**Table 1 healthcare-12-00389-t001:** Research indicators, descriptive, and dorrelations.

	Consenting to Unwanted Sex	Self-Esteem	Depression	Sexual Abuse	Age	Education
Withholding Sex (*Mean* = 0.543 *SE* = 0.048)	0.540 ***	−0.173 **	0.268 ***	n.s	0.158 *	n.s
Consenting to Unwanted Sex (*Mean* = 0.947 *SE* = 0.053)		−0.331 ***	0.295 ***	n.s	n.s	−0.139 *
Self-Esteem (*Mean* = 3.096 *SE* = 0.033)			−0.564 ***	−0.215 **	0.279 ***	0.214 ***
Depression (*Mean* = 2.463 *SE* = 0.047)				0.227 **	−0.295 ***	−0.254 ***
Sexual Abuse (*Mean* = 4.048 *SE* = 0.230)					n.s	n.s
Age (*Mean* = 27.979 *SE* = 0.516)						0.361 ***
Education (*Mean* = 2.831 *SE* = 0.042)						

N = 236. *p* > 0.05 = n.s; * *p* < 0.05; ** *p* < 0.01; *** *p* < 0.001.

## Data Availability

Data are contained within the article.

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
