# Peer review of "The Association between Women’s History of Sexual Abuse, Mental Health, and Sexual Motivations in Committed Intimate Relationships"

_healthcare, 2024, doi:10.3390/healthcare12030389_

Round 1
Reviewer 1 Report
Comments and Suggestions for Authors
- Very interesting topic
- Needs another run through for grammar top to bottom (for example line 93)
- Kudos for including sample size estimations
- In the limitations, around line 418, the authors begin an important point that I think should be developed further, and with perhaps a few other citations, or additional context. By that what I'm referring to is not only that some people recover from the trauma initiated by abuse, but there's also an important body of research that exists related to healthy feminine sex drives in general. Perhaps just a bit in this space to state how the project was specifically not looking into feminine sexual desire, but rather negative reactions to trauma that manifest (or don't) in sexual relationships. I think the authors are on the cusp of it, but a large segment of your potential readership could appreciate the acknowledgement of women as sexual beings in a more well rounded way.
- I think this concept expands to the entire article, which I would go through and look for ways to adjust language here and there to adjust for pro-feminine sexuality and empowerment. There's a great deal of research that says women like sex, if not that women are the anthropological gatekeepers of physical intimacy. I know thats not what the core of the article is about, but calling that in, in small ways, could have a big impact on readership adoption.
- The science of the manuscript is sound. Well done.
Comments on the Quality of English LanguageNeeds minor edits here and there.
Author Response
We thank Reviewer 1 for their helpful comments!
Reviewer 1:
- Very interesting topic
Response: Thank you!
- Needs another run through for grammar top to bottom (for example line 93)
Response: We carefully read throughout the manuscript to capture and correct grammatical mistakes.
- Kudos for including sample size estimations.
Response: Thank you!
- In the limitations, around line 418, the authors begin an important point that I think should be developed further, and with perhaps a few other citations, or additional context. By that what I'm referring to is not only that some people recover from the trauma initiated by abuse, but there's also an important body of research that exists related to healthy feminine sex drives in general. Perhaps just a bit in this space to state how the project was specifically not looking into feminine sexual desire, but rather negative reactions to trauma that manifest (or don't) in sexual relationships. I think the authors are on the cusp of it, but a large segment of your potential readership could appreciate the acknowledgement of women as sexual beings in a more well rounded way.
I think this concept expands to the entire article, which I would go through and look for ways to adjust language here and there to adjust for pro-feminine sexuality and empowerment. There's a great deal of research that says women like sex, if not that women are the anthropological gatekeepers of physical intimacy. I know thats not what the core of the article is about, but calling that in, in small ways, could have a big impact on readership adoption.
Response: Thank you for this important comment. We have added text in the Limitations section to reflect this idea more, including further citations to support this assertion.
- The science of the manuscript is sound. Well done.
Response: Thank you!
Comments on the Quality of English Language
Needs minor edits here and there.
Response: Done.
Reviewer 2 Report
Comments and Suggestions for Authors
1

Author Response
Reviewer 2:
The introduction is very clear; The concepts with which we are going to work are adequately explained (mental health and motivations towards intimate relationships), but I miss a section in which the concept of sexual abuse with which we are going to work is conceptualized. It must be considered that sexual abuse can be understood in different ways according to the legislation of each country. For example, in Spain, the legislation was recently changed regarding what is considered sexual abuse. This concept is therefore variable and it is necessary to indicate the parameters that will be used in this study. Therefore, authors should include a section that makes it clear what they will consider sexual abuse.
Response: Thank you for this important suggestion. We have added a section (#1.3) where we define and discuss sexual abuse in the Introduction.
The hypotheses are clear and founded.
Response: Thank you!
The procedure is appropriate.
Response: Thank you!
Sample: technically couples who do not live together under the same roof, despite their commitment, are considered dating relationships, and therefore the study conditions of this type of couples are different from those of studies of couples who live together. For example, in studies of dating violence, both types of couples are studied differently. In this work, 30.9% of the sample are in dating relationships, therefore, the authors must justify why these relationships can be studied with the same parameters and without distinction with couples who live together under the same roof. Since in the literature they are studied separately.
Response: Thank you for this comment. Although we understand the reviewer's question, we wanted to allow the sample to include couples in different statuses. The focus for us was that the participants had to be involved in a committed intimate relationship for at least a year. We added text under Participants to explain that in order to be included in this study, participants had to be involved in what they defined a committing exclusive relationship. In addition, we ran the model with relationship status to explore its potential associations with the study's variables, and results indicated no significant associations.
Very interesting and positive about the exhibition is the fact that religion is taken into consideration.
Response: Thank you!
Instruments: totally appropriate and justified.
Response: Thank you!
Results and Discussion: clearly explained and substantiated
Response: Thank you!

Reviewer 3 Report
Comments and Suggestions for Authors
11. On page 4, third paragraph and 5th paragraph, “α Cronbach” should be replaced by “Cronbach’s α”.
22. On page 4, the end of 5th paragraph, “rp=-.56” is only a moderate correlation, not a strong one.
33. At top of page 6, “rp=.54” is only a moderate correlation, not a strong one.
44. On page 7, middle of first paragraph, does “p>.05” mean that the model is not statistically significant?
55. On page 10, first paragraph, the authors will add “the small sample size” as another limitation of this study.
Author Response
Reviewer 3:
- On page 4, third paragraph and 5thparagraph, “α Cronbach” should be replaced by “Cronbach’s α”.
Response: Thank you. We made the changes.
- On page 4, the end of 5thparagraph, “rp=-.56” is only a moderate correlation, not a strong one.
Response: Thank you for your comment. According to relevant literature in psychology and social sciences (e.g., Coolican, 2017; Fitz et al., 2012; Labakov & Agadullina, 2021), Pearson's effect size r is considered strong at 0.5 and above. Thus, we describe these effect sizes as strong.
- At top of page 6, “rp=.54” is only a moderate correlation, not a strong one.
Response: Please see our previous response.
- On page 7, middle of first paragraph, does “p>.05” mean that the model is not statistically significant?
Response: Thank you very much for capturing this. This is a typo. We corrected this both on p. 7 and in Figure 1.
- On page 10, first paragraph, the authors will add “the small sample size” as another limitation of this study.
Response: We added this as another limitation of the study.
References
Coolican, H. (2017). Research methods and statistics in psychology. Psychology press.
Fritz, C. O., Morris, P. E., & Richler, J. J. (2012). Effect size estimates: current use, calculations, and interpretation. Journal of Experimental Psychology: General, 141(1), 2-18.
Lovakov, A., & Agadullina, E. R. (2021). Empirically derived guidelines for effect size interpretation in social psychology. European Journal of Social Psychology, 51(3), 485-504.
